# Viability of Red (*Ribes rubrum* L.) and Black (*Ribes nigrum* L.) Currant Cuttings in Field Conditions after Cryopreservation in Vapors of Liquid Nitrogen

**Vladimir Verzhuk *** , **Alexander Pavlov, Liubov Yu. Novikova** and **Galina Filipenko**

Federal Research Center, The N.I. Vavilov All-Russian Institute of Plant Genetic Resources, 42, 44 Bolshaya Morskaya st., 190000 St. Petersburg, Russia; pavlov-al@bk.ru (A.P.); l.novikova@vir.nw.ru (L.Y.N.); g.filipenko@vir.nw.ru (G.F.)

*   Correspondence: vverzhuk@mail.ru

**Abstract:** One of the prospective ways to safely preserve the genetic resources of red and black currant for breeding needs is the cryopreservation of cuttings with dormant buds. Vegetative cuttings of 12 varieties of red and 11 varieties of black currant were harvested in various regions of Russia: North, Northwest, and Central. Their viability after cryopreservation in nitrogen vapors (about −184 °C) under field conditions was studied. For red currant samples, it ranged from 61.2 ± 1.2% to 72.3 ± 3.0%, black currant—from 58.9 ± 1.1% to 73.5 ± 1.9%. In the group of red currant varieties, there were no significant differences in viability between varieties after cryopreservation. In the group of black currant varieties, "Chereshneva" and "Georgiy" had lower viability after storage in liquid nitrogen vapors than the others, 61.1% and 58.9%, respectively. On red currant, dry growing conditions of the experiment year significantly decreased the viability after cryopreservation. Neither black nor red currants revealed the influence of the place of harvesting on the survival of cuttings after cryopreservation. These results indicate the possibility of using cryopreservation to preserve cuttings of red and black currant with dormant buds collected in regions with different climatic environments.

**Keywords:** cryopreservation; plant genetic resources; berry crops; liquid nitrogen vapor

## 1. Introduction

Due to the instability of climatic, environmental, and economic conditions in the world, there is a threat of irreplaceable loss of valuable collection samples of vegetative propagated fruit and berry crops. Preservation of biological diversity is possible by storage in situ, i.e., conditions where genetic resources exist within ecosystems and natural habitats, and ex situ—outside natural habitats. Cryopreservation of plant parts (cuttings, buds, pollen, and meristems) in liquid nitrogen (−196 °C) or its vapor (about −184 °C) is a promising method for storing the gene pool of fruit crops ex situ. At this temperature, all metabolic processes practically stop, due to which living objects can be preserved for a long time [1]. Cryopreserved tissues and cells may be placed in cryogenic storage via two main routes, involving either direct immersion into liquid nitrogen (nitrogen vapors) or control rate (programmable) cooling [2].

Sakai [3] provided one of the first studies showing that winter twigs of poplar (*Populus sieboldi*) and willow (*Salix koriyanagi*) could survive ultralow temperatures if slowly cooled to −30 °C or lower prior to immersion in liquid nitrogen. It is even possible to carry out cryopreservation in liquid oxygen. A later study demonstrated that this simple methodology was also applicable to twigs of several fruit species [4–8]. Earlier, Tumanov et al. [9] found that shoots of black currant and birch can successfully survive cooling down to −254 °C (temperature of liquid hydrogen) if they are slowly cooled beforehand to −60 °C.

Cryopreservation using a winter vegetative bud method is being applied to the *Malus* collection maintained in the field at the USDA-ARS Plant Genetic Resources Unit, Geneva, New York [10–14], and to *Malus* accessions from the Canadian Clonal Genebank of Plant Gene Resources of Canada [15]. This method is used to preserve accessions from clonal wild species *Malus* collection at the Julius Kühn-Institute, Institute for Breeding Research on Fruit Crops, Germany [16].

N.I. Vavilov All-Russian Institute of Plant Genetic Resources (VIR) has developed methods for cryopreservation of cuttings with dormant buds of many crops with preliminary cooling to temperatures −30 or −50 °C: apricot, bird cherry, cherry, cherry plum, gooseberry, pear, plum, sweet cherry [17–20]. In recent years, work has been carried out with black and red currants. The main task of this study was to evaluate the cutting viability of black and red currant after cryopreservation in the field. Since the degree of cold hardiness is known to be a factor in the successful use of dormant bud cryopreservation, it was interesting to find out whether material from a relatively warmer climate would respond worse to cryopreservation than material from a colder environment.

## 2. Materials and Methods

Twelve red currant and eleven black currant cultivars served as the material for the research (Table 1). These cultivars have different ripening periods and belong to different regions of cultivation [21–26].

**Table 1.** Red and black currant accessions selected for six-month storage under low and ultralow temperatures. Cuttings were collected during 2017–2019.

| No. | Name of the Cultivar | Catalogue Number | Collecting Site |
|-----|----------------------|------------------|-----------------|
| **Red Currant Accessions (*Ribes rubrum* L)** | | | |
| 1 | Laplandiya | k-315 | |
| 2 | Natali | k-202 | |
| 3 | Svetlana | k-201 | Polar Experiment Station of VIR (Apatity) |
| 4 | Tatyana | k-313 | |
| 5 | Zarya Zapolyarya | k-200 | |
| 6 | Asora | – | |
| 7 | Asya | – | |
| 8 | Belka | – | |
| 9 | Marmeladnitsa | – | I.V. Michurin FSC (Michurinsk) |
| 10 | Ogonyok | – | |
| 11 | Osipovskaya | – | |
| 12 | Vika | – | |
| **Black Currant Accessions (*Ribes nigrum* L.)** | | | |
| 1 | Andreyevskaya | k-15630A | |
| 2 | Chereshneva | k-42481 | |
| 3 | Dumushka | k-42516 | Pushkin & Pavlovsk Labs of VIR (Pavlovsk) |
| 4 | Fat | k-42509 | |
| 5 | Venera | k-42522 | |
| 6 | Georgiy | – | |
| 7 | Dachnitsa | – | |
| 8 | Ekzotika | – | I.V. Michurin FSC (Michurinsk) |
| 9 | Krestetskaya | – | |
| 10 | Selechenskaya | – | |
| 11 | Tambovskaya pozdnyaya | – | |

The yearly growth of sprouts was selected in the phase of plant dormancy (November to December) in different regions of Russia: the collection plantations of the Polar Experiment Station of VIR (Apatity),

I.V. Michurin Federal Scientific Center (Michurinsk), and Pushkin and Pavlovsk Laboratories of VIR (Pavlovsk) (Table 2).

**Table 2.** Geographical coordinates and altitudes of harvest locations.

| Location | Latitude | Longitude | Elevation (m) |
|---|---|---|---|
| Apatity | 67°34′ N | 33°24′ E | 170 |
| Michurinsk | 52°53′ N | 40°29′ E | 150 |
| Pavlovsk | 59°44′ N | 30°23′ E | 74 |

Then, in the VIR laboratory for long-term storage of plant genetic resources, the cuttings were divided into segments 7–8 cm long, with 2–3 buds in a segment. The initial viability of the collected material (control in our experiment) was assessed by growing 10 cuttings, with three replications per each cultivar in the glass containers with water, under 21 ± 1 °C, 16 h light/8 h dark, until the formation of leaves and roots. A part of the cuttings was left as reference (when transplanting them in spring into the soil) and stored in a HUURRE refrigerator at −5 °C, while the larger part of the plant material was dried at −4 °C down to the required moisture in the plants, 28–32%. After drying, the cuttings were gradually frozen in foil laminated packages using a multistep technique. Freezing to −30 °C was carried out at a rate of 0.5 °C per min. At −30 °C, the cuttings were kept for 30 min. Then the cuttings were frozen to a temperature from −48 to −50 °C at a rate of 1 °C per min. Then, the frozen samples were placed into cryopreservation tanks for long-term storage in liquid nitrogen vapor at a temperature from −183 to −185 °C for six months. In the spring, the cuttings were removed from the tanks, defrosted in cold water, and transplanted in the field at Pavlovsk to assess their viability. At the same time, cuttings were planted in the field, which were stored in the refrigerator at −5 °C. The viability of both frozen and refrigerated cuttings in the field was assessed by growing 10 cuttings with three replicates for each cultivar. In the spring/summer growing season, morphological observations were performed to analyze the establishment of various red and black currant cultivars in the field and monitor the growth and development of plants after the stress storage conditions.

The growing conditions of samples at their harvest points in 2016–2018 were different by temperature and precipitation during the period, with temperatures above 10 °C: in the Polar Experiment Station of VIR, the sum of daily temperatures at this period was 1067 °C, precipitation 93 mm; in Pavlovsk 2435 °C and 440 mm; and in Michurinsk 2913 °C and 345 mm (Figure 1).

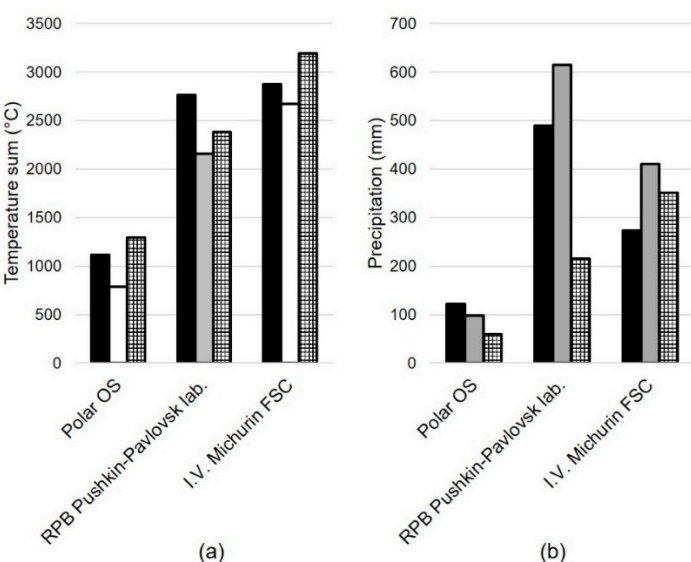

**Figure 1.** Plant vegetation conditions prior to selection of cuttings: (**a**) temperature sum and (**b**) precipitation.

In Pavlovsk, where the viability of cuttings after cryopreservation in the field was determined, 2017 and 2018 during the period of planting and survival of cuttings were not quite favorable in terms of precipitation, but in the following months, it was enough. In 2019, from the middle of the third decade of May (cuttings were planted in the field on 24 May) to September, dry and windy weather persisted. The warmest year was 2018, with the average monthly temperature reaching 19.2 °C in July, and the maximum daily temperature reaching 30 °C. The coldest was 2019, when the warmest month was June, 17.7 °C, and in July and August the temperatures were 15.8 and 15.4 °C. The wettest year was 2017 with a total of 535 mm of precipitation for June–August, and the least precipitation occurred in 2019, 117 mm (Figure 2). Therefore, better survival of the transplanted cuttings in 2019 required frequent watering in spring and summer, so that the survived plants in the field did not dry out and die.

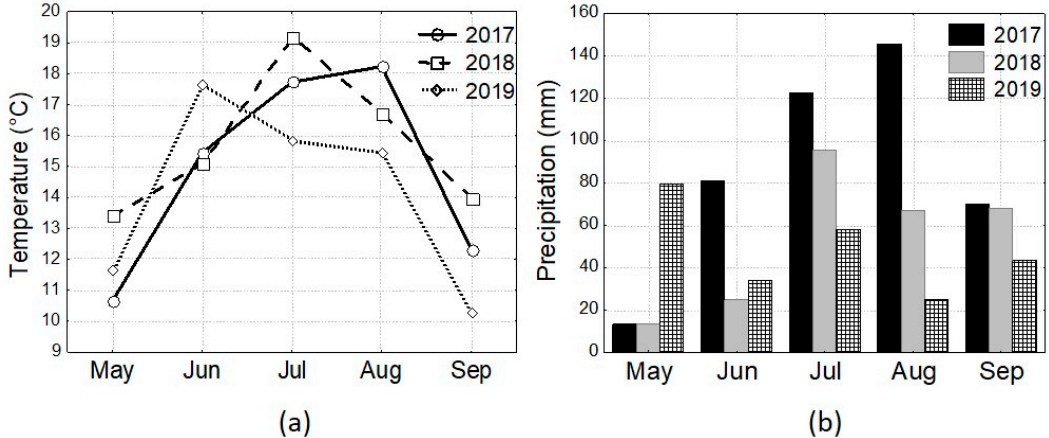

**Figure 2.** Weather conditions of the field experiment at the RPB Pushkin–Pavlovsk Laboratories of the Vavilov All-Russian Institute of Plant Genetic Resources (VIR) for determining the viability of cuttings: (**a**) average monthly temperatures and (**b**) monthly total precipitation.

The significance of the collecting site and year was investigated by two-way ANOVA in the Statistica 13.3 package (TIBCO Software Inc., Palo Alto, CA, USA); a post hoc analysis was performed by using the LSD test [27]. Difference between cultivars was tested by one-way ANOVA. The significance of differences in storage methods and between years was clarified using Student's *t*-test for dependent samples. The significance level is 5%.

## 3. Results and Discussion

Table 3 presents the results of a three-year experiment to determine the viability of red currant cuttings before and after cryopreservation in liquid nitrogen vapor. The initial viability was determined in the fall under laboratory conditions. The viability of cuttings after cryopreservation was assessed in the field.

The initial viability of the red currant cuttings ranged from 85.6 ± 4.3% to 93.3 ± 3.3%. The viability of the red currant cuttings after cryopreservation, measured in the field, varied from 58.9 ± 2.0 to 72.3 ± 3.2%. This is above 40% viability—the minimum requirement for samples to be laid for long-term cryogenic storage [12].

**Table 3.** The effect of low-temperature storage and cryopreservation on the viability of red currant cuttings selected in different regions of Russia (2017–2019).

| No. | Cultivar | Catalogue No. | Viability of the Cuttings (%) | | |
|---|---|---|---|---|---|
| | | | Initial | After Storage under −5 °C | After Cryostorage under −183 to −185 °C |
| **Cuttings from Polar Experiment Station of VIR, town of Apatity** | | | | | |
| 1 | Laplandiya | k-315 | 90.0 ± 2.5 | 75.6 ± 1.1 | 70.2 ± 3.1 |
| 2 | Natali | k-202 | 85.6 ± 5.6 | 71.2 ± 4.1 | 63.4 ± 2.0 |
| 3 | Svetlana | k-201 | 88.9 ± 1.1 | 72.2 ± 1.1 | 62.3 ± 1.1 |
| 4 | Tatyana | k-313 | 93.3 ± 0.2 | 74.4 ± 1.1 | 68.9 ± 1.1 |
| 5 | Zarya Zapolyarya | k-200 | 88.9 ± 1.1 | 72.3 ± 5.9 | 66.7 ± 1.9 |
| | Mean value | | 89.3 ± 1.2 | 73.1 ± 0.8 | 66.3 ± 1.5 |
| **Cuttings from Michurin Federal Scientific Center, town of Michurinsk** | | | | | |
| 1 | Asora | – | 90.5 ± 2.4 | 78.9 ± 1.1 | 67.9 ± 2.9 |
| 2 | Asya | – | 85.6 ± 1.1 | 79.1 ± 2.7 | 70.1 ± 1.7 |
| 3 | Belka | – | 90.5 ± 2.4 | 78.9 ± 1.1 | 72.3 ± 3.0 |
| 4 | Marmeladnitsa | – | 86.7 ± 1.9 | 74.6 ± 1.1 | 70.0 ± 1.9 |
| 5 | Ogonyok | – | 86.7 ± 1.9 | 70.1 ± 1.8 | 61.2 ± 1.2 |
| 6 | Osipovskaya | – | 93.4 ± 3.3 | 79.9 ± 1.8 | 71.2 ± 4.1 |
| 7 | Vika | – | 89.9 ± 1.8 | 74.4 ± 1.1 | 68.9 ± 1.1 |
| | Mean value | | 89.0 ± 1.1 | 76.6 ± 1.4 | 68.8 ± 1.4 |

However, since freezing/unfreezing of biological objects is a physiologically traumatic procedure, the viability of cuttings after six months of storage under low temperatures in our experiment was somewhat higher than after cryopreservation. Thus, the average viability of the cuttings taken from the collection of the Polar Experiment Station after cryopreservation was 66.3%, which is lower than 73.1% after storage at −5 °C (significance level of differences according to Student's *t*-test for dependent samples at $p = 0.002$). The mean viability of the cuttings from the Michurin Federal Scientific Center (Michurinsk) after cryopreservation was 68.8%, and after storage under low temperatures, 76.6% ($p < 0.001$). For both sites of selection, the mean viability of the cuttings after cryogenic storage (67.8%) was significantly lower than after low-temperature storage (75.1%) ($p < 0.001$).

It should be mentioned that low-temperature storage of fruit and berry crop cuttings is usually applied for one or two years. Afterwards, the stored plant material often dies, since this type of storage does not completely block the metabolic processes in the cuttings' tissues, which leads to a rapid decrease in the viability of cuttings. In addition, it is difficult to provide an optimal level of humidity under such conditions for a long time; therefore, the fungal and bacterial microflora that inevitably presents in plant tissues is manifested. A large role is played by the fact that woody plants consist of tissues with varying resistance to low temperatures, the upper third of the shoots is not mature enough, since the end of visible growth is not an indicator of the end of vegetation in plants [28]. It is assumed that these problems do not appear in ultralow temperatures [29–31]. The continuation of our experiment will allow us to clarify these provisions.

One-way analysis of variance showed that among the varieties of Polar OS (Apatity) after storage at −5 °C there are no significant differences ($p = 0.875$) in viability, and after cryostorage ($p = 0.076$) there are no significant differences either. Among the group of varieties from I.V. Michurin FSC (Michurinsk), significant differences were observed at −5 °C ($p = 0.007$), but not after cryostorage ($p = 0.108$). A post hoc analysis showed a significant difference after −5 °C between "Osipovskaya" varieties with the highest viability of 79.9% and "Ogonyok" with the lowest at 70.1%.

Figure 3 shows the average viability of red currant cuttings (initial and after six-month storage at −5 and −184 °C) for different collection sites in different years of research.

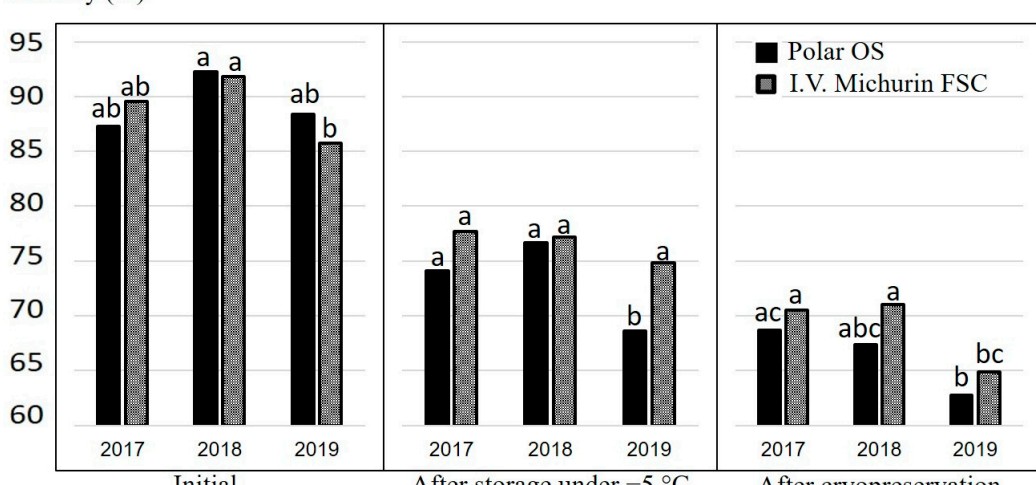

**Figure 3.** Viability of red currant cuttings collected in Polar OS and I.V. Michurin FSC after storage. Within each box, bars with different letters differ for $p < 0.05$.

For red currants, two-way ANOVA showed that the place of origin did not affect the initial viability of cuttings ($p = 0.824$), the year of cultivation had a significant effect ($p = 0.015$), and the combined effect of factors was insignificant ($p = 0.345$). Student's *t*-test for dependent samples showed that for varieties from the Polar OS there are no significant differences between years (87.3% in 2017, 92.2% in 2018, 88.4% in 2019; 2017–2018 $p = 0.233$; 2017–2019 $p = 0.173$; 2018–2019 $p = 0.347$); for varieties from Michurinsk, 2019 was significantly worse than 2017 and 2018 (89.5% in 2017, 91.8% in 2018, 85.7% in 2019; 2017–2018 $p = 0.085$; 2017–2019 $p = 0.049$; 2018–2019 $p = 0.003$).

The viability of cuttings after −5 °C was affected by both the place of origin ($p = 0.021$) and the year of study ($p = 0.012$); the combined influence of factors was insignificant ($p = 0.275$). Similar results were obtained by Jenderek et al. 2011 [32], who were engaged in cryopreservation of *Malus* L. cuttings grown in different places in the USA (Geneva, NY; Corvallis, OR; Davis, CA). Varieties from Michurinsk were characterized by a higher viability of 76.6% than from Polar Station 73.1%. For varieties from the Polar Station, 2017–2019 differed significantly, 2019 was characterized by the worst viability (74.1% in 2017, 76.7% in 2018, 68.6% in 2019; 2017–2018 $p = 0.412$; 2017–2019 $p = 0.018$; 2018–2019 $p = 0.099$); for varieties from Michurinsk, 2018 and 2019 differed significantly, 2019 was the worst (77.7% in 2017, 77.2% in 2018, 74.8% in 2019; 2017–2018 $p = 0.788$; 2017–2019 $p = 0.118$; 2018–2019 $p = 0.006$).

After cryopreservation, there was also no influence of the place of origin ($p = 0.077$), but the year was significant ($p = 0.003$), and the combined influence of factors was insignificant ($p = 0.844$). For Polar Station, the lowest viability was in 2019 (68.7% in 2017, 67.4% in 2018, 62.8% in 2019; 2017–2018 $p = 0.196$; 2017–2019 $p = 0.007$; 2018–2019 $p = 0.019$), and similarly for varieties from Michurinsk (70.5% in 2017, 71.0% in 2018, 64.9% in 2019; 2017–2018 $p = 0.750$; 2017–2019 $p = 0.008$; 2018–2019 $p = 0.008$).

Table 4 presents the results of a three-year experiment to determine the viability of black currant cuttings before cryopreservation, after cryopreservation in nitrogen vapor, and at −5 °C in the refrigerator.

Testing the initial viability of black currant cuttings showed that their viability after germination in water in the light room was within the range 83.3 ± 3.3% to 94.4 ± 1.1%. The viability of black currant cuttings after cryopreservation, determined in the field, ranged from 58.9 ± 1.1% to 73.5 ± 1.9%.

**Table 4.** The effect of low-temperature storage and cryopreservation on the viability of black currant cuttings selected in different regions of Russia (2017–2019).

| No. | Cultivar | Catalogue No. | Viability of the Cuttings (%) | | |
|---|---|---|---|---|---|
| | | | Initial | After Storage under −5 °C | After Cryostorage under −183 to −185 °C |
| **Cuttings from Pushkin and Pavlovsk Laboratories of VIR, town of Pavlovsk** | | | | | |
| 1 | Andreyevskaya | k-15630A; | 93.3 ± 0.2 | 83.3 ± 1.9 | 72.2 ± 1.1 |
| 2 | Chereshneva | k-42481 | 83.3 ± 3.3 | 76.7 ± 1.9 | 61.1 ± 1.1 |
| 3 | Dumushka | k-42516 | 86.6 ± 1.9 | 80 ± 1.9 | 68.9 ± 1.1 |
| 4 | Fat | k-42509 | 86.7 ± 1.9 | 80.1 ± 2 | 67.8 ± 2.9 |
| 5 | Venera | k-42522 | 89.7 ± 2.2 | 82.2 ± 1.1 | 71.2 ± 1.2 |
| | Mean value | | 87.9 ± 1.7 | 80.5 ± 1.1 | 68.2 ± 2 |
| **Cuttings from Michurin Federal Scientific Center, town of Michurinsk** | | | | | |
| 1 | Georgiy | – | 85.6 ± 1.1 | 70.1 ± 1.8 | 58.9 ± 1.1 |
| 2 | Dachnitsa | – | 94.4 ± 1.1 | 82.3 ± 1.2 | 73.5 ± 1.9 |
| 3 | Ekzotika | – | 86.7 ± 1.9 | 80.1 ± 2 | 70.2 ± 3.1 |
| 4 | Krestetskaya | – | 89.6 ± 1.6 | 75.6 ± 1.1 | 67.9 ± 2.9 |
| 5 | Selechenskaya | – | 91.1 ± 1.1 | 82.3 ± 1.2 | 73.5 ± 1.9 |
| 6 | Tambovskaya pozdnyaya | – | 87.8 ± 1.1 | 80 ± 0 | 66.8 ± 1.8 |
| | Mean value | | 89.2 ± 1.3 | 78.4 ± 1.9 | 68.5 ± 2.2 |

A comparison of the viability of cuttings planted in the field after storage at −5 °C and after storage in nitrogen vapor showed that the varieties taken in the collection area of Pavlovsk had a lower average viability of cuttings after cryopreservation (68.2%) than after low-temperature storage (80.5%) ($p < 0.001$). Cuttings cut in the collection garden in Michurinsk also had less viability after cryopreservation (68.5%) than after storage at −5 °C (78.4%) ($p < 0.001$). The average viability of cuttings after cryopreservation (68.4%) was significantly lower than after low-temperature storage (79.3%) ($p < 0.001$).

There were no significant differences at −5 °C ($p = 0.171$) among the varieties from the Pushkin and Pavlovsk laboratory, but there were after cryopreservation ($p = 0.006$); the "Chereshneva" variety with the lowest survival rate of 61.1% was distinguished. Among the varieties from the I.V. Michurin Federal Scientific Center, significant differences were noted after −5 °C ($p < 0.001$); the lowest viability was distinguished in varieties "Georgiy" (70,1%) and "Krestetskaya" (75.6%). There were also significant differences after cryopreservation ($p = 0.006$); the least viable variety was also distinguished by "Georgiy" (58.9%).

Figure 4 shows the average viability of black currant cuttings (initial and after storage for six months at −5 and −184 °C) for different collection sites in different years of research.

For black currant, no effect of the place of harvest or the year of study was found for either the control group or for samples after low-temperature storage, or for samples after cryopreservation by both two-way analysis of variance and Student's *t*-test for dependent samples.

In all the years of research, plants during the spring and summer vegetation formed two or three young shoots from the buds, prepared for overwintering. In 2019, it was noted that plants grown after storage in the refrigerator at −5 °C were affected by powdery mildew, while those stored in nitrogen vapor did not suffer from this disease. It is planned to continue work on studying the effect of cryopreservation on the resistance of currants to bacterial and viral diseases.

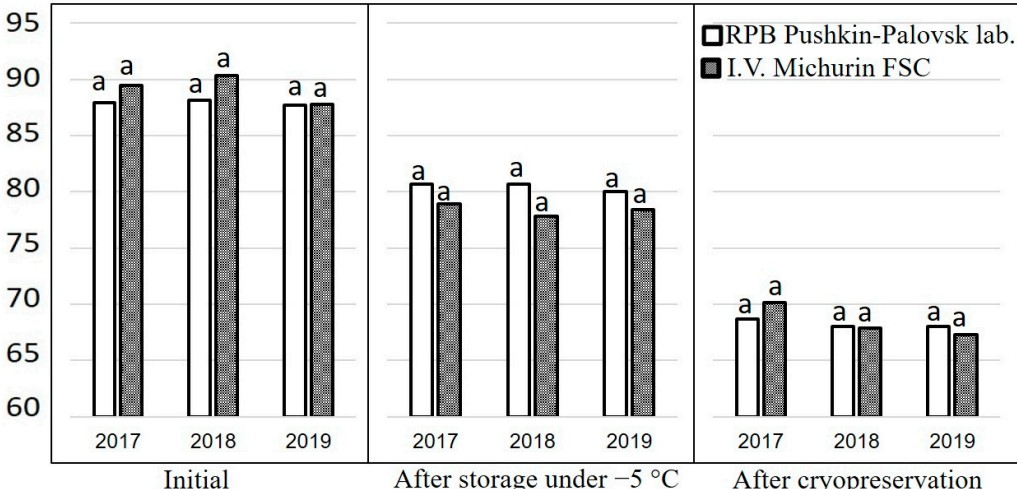

**Figure 4.** Viability of black currant cuttings collected at the Research and Production Base Pushkin–Pavlovsk Laboratories of VIR and I.V. Michurin FSC after storage. Within each box, bars with different letters differ for $p < 0.05$.

## 4. Conclusions

According to the results of the studies, it was noted that the viability of currant cuttings planted in the field after storage in liquid nitrogen vapor was high, and its values for the red currant were in the range from $61.2 \pm 1.2\%$ to $72.3 \pm 3.0\%$, for black currant—from $58.9 \pm 1.1\%$ to $73.5 \pm 1.9\%$. This is exceeds the minimum requirement for samples to be laid for long-term cryogenic storage.

On red currants, the effect of environment in the year of study on the level of viability after cryopreservation was found; the dry growing conditions of 2019 significantly decreased the viability after cryopreservation. On black currant, the existence of varietal differences in the ability of cuttings to survive ultralow temperatures was shown.

The effect of the place of harvest was not found to influence the viability of cuttings after cryopreservation and their survival in the field, neither for black nor red currants. These results indicate the possibility of using the control rate cooling to preserve cuttings of red and black currant with dormant buds collected in regions with different climatic environments.

Keeping cuttings of red and black currants at $-5\,°C$ can be used for short-term conservation of the genetic resources of these crops.

**Author Contributions:** Conceptualization: V.V.; experiment performance; V.V. and A.P.; data analysis and visualization: L.Y.N.; writing—original draft preparation; V.V., A.P., and G.F.; writing—review and editing; G.F. All authors have read and agreed to the published version of the manuscript.

**Funding:** This research received no external funding.

**Acknowledgments:** The work was carried out within the framework of the state task according to the VIR thematic plan on the topic No. 0662-2019-0004 "Collections of vegetatively propagated crops (potatoes, fruit, berry, grapes) and their wild relatives—study and rational use".

**Conflicts of Interest:** The authors declare no conflict of interest.

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
