# Peer review of "Viability of Red (Ribes rubrum L.) and Black (Ribes nigrum L.) Currant Cuttings in Field Conditions after Cryopreservation in Vapors of Liquid Nitrogen"

_agriculture, doi:10.3390/agriculture10100476_

Round 1
Reviewer 1 Report
This study aimed to determine the effect of cryopreservation in vapors of liquid nitrogen on the survival of black and red currants. The results show that the cryopreservation in vapors of liquid nitrogen decreases the survival of the buds. The survival after cryopreservation is worse than after storage at -5 ° С. However, according to the authors, the rate of survival after cryopreservation in vapors of liquid nitrogen is above the minimum requirement for samples to be laid for long-term cryogenic storage, and the authors claimed that cryopreservation in vapors of liquid nitrogen protected plants from powdery mildew (data not shown). The manuscript is generally written in an understandable manner, although I think that the following points should be improved.
- Title: “Cryopreservaition” should be replaced by “Cryopreservation”
- “red currant” or “redcurrant”? : For homogeneity reasons, the authors should choose the same version of writing in all the manuscript I think. Same comment for “black currant” or “blackcurrant”.
- For a better visibility, the results of the statistics should be added to the tables and figures as well I think (different letters could show differences).
- Line 205: “It should also be noted that plants grown after storage in nitrogen vapor, during the spring and summer vegetation were not susceptible to various diseases and pests, compared to those stored in the refrigerator at –5°C, which suffered such a disease as powdery mildew”. If the authors collected some data about diseases and pests, the results should be presented in the manuscript. I think it would add a great value to the manuscript.
- Lines 34-35: “Cryopreservated tissues and cells may be placed in cryogenic storage two main routes,” please revise the writing.
- Line 49: please indicate here the full name for “VIR”
- Line 69: please indicate the meaning of “PGR”. All abbreviations should be explained.
- Lines 70-72 mention that the viability was checked in vitro, while lines 80-82 mention that the viability was checked in the field. Please indicate that the viability in vitro was done as a control, and how many replicates were used for the field experiment. This section should also mention that the viability was checked after storage at - 5°C and how many replicates were used for that control.
- Lines 87-88: please precise how the sum of temperatures was calculated.
- Line 127: “storage at –5°C –73.1%” please correct the writing.
- Lines 158-159: “for varieties from Michurinsk, 2019 was significantly the worst”: please elaborate.
Reviewer 2 Report
Dear editor and dear colleagues,
I have read the submitted research article “Viability of Red (Ribes rubrum L.) and Black (Ribes nigrum L.) Currant Cuttings in the Field Conditions after Cryopreservation in Vapors of Liquid Nitrogen” with great interest.
It is an article that deals with the cryopreservation of an important fruit of the temperate climate (red/black Currans). It is a very interesting subject that fits well within the scope of this journal and provide some caveats are better explained or resolved.
Major remarks
My major concern is that the authors have used a statistical methodology and an experimental design that does not fully exploits/enables direct comparison and interactions display across independent/dependent variables, as supposed to. There are no common cultivars across the collection areas; hence safe conclusions cannot be drawn for genotype X area interactions and possible viability bias. Different cultivars were used at different environments; hence a MAVOVA analysis cannot be employed. t-test, one-way anova and two-way anova (used in the current version) do not fully reflect the optimal experimental design in order to identify possible correlations.
There are (should be) concurrently four independent variables.
1. Location: Apatity, Michurinsk and Pavlovsk (a nominal character),
2. Cultivars (also a nominal character),
3. Type of currant (red or black - also a nominal character) and
4. The year of collection (also a nominal character).
The dependent variables (all scale characters) are the percentage of viability under three treatments:
1. Control,
2. -5°C and
3. Liquid nitrogen vapors
As a result, this should be a multivariate experimental design; unfortunately, there are several ‘gaps’ in the present form. I strongly urge the authors to provide the basis and rational for this ’truncated’ design.
Some minor points
The article could use proofreading from an English language expert, since there are scattered grammatical and syntax errors throughout the manuscript (it is not in the scope of a scientific review to demarcate them all - but rather to highlight a few as, an example)
To name some:
When using a nomenclature in Latin (ie in situ, ex situ) always use italics.
The formatting of temperature variation using suspension points/ellipsis (9–4°…–5°Ð¡) is not very straightforward. I can understand that since there are minus Celsius temperatures a dash (–) cannot be easily used. Better to adopt a format like: temperature varying from -183°Ð¡ to -185°Ð¡, (183°Ð¡ to -185°Ð¡), circa -185°Ð¡, about -185°Ð¡ etc.
L 4: Cryopreservaition. Change to Cryopreservation
L18-19: On red currant, the effect of growing conditions of experiment year on the level of viability after cryopreservation was found.
What was the effect?
L35: Cryopreservated tissues and cells may be placed in cryogenic storage two main routes,
Rephrase to: Cryopreservated tissues and cells may be placed in cryogenic storage via two main routes
L71: with three replications per each cultivar in the glasses with water in a lighted room under 21º ± 1ºÐ¡, 16 hours per day, and 8 hours per night
You mean glass containers? 16 h light/8 h dark?
Based on the above reason a recommend a major revision
Round 2
Reviewer 2 Report
Dear editor and authors,
I have concluded my review of the revised paper and i feel that the manuscript has been improved.
Despite the fact that the choice of genotypes does not fully reach the optimal experimental design, still i can fully understand the limitations of Currant cultivation to such diverse environmental conditions.
As a result, the majority of the concerns raised has been adequately addressed and my suggestion is to accept this manuscript for publication